# TV News and COVID-19: Media Influence on Healthy Behavior in Public Spaces

**DOI:** 10.3390/ijerph18041879

**Published:** 2021-02-15

**Authors:** Massimiliano Scopelliti, Maria Giuseppina Pacilli, Antonio Aquino

**Affiliations:** 1Department of Human Studies, Libera Università Maria Ss. Assunta (LUMSA University), 00193 Roma, Italy; 2Dipartimento di Scienze politiche, University of Perugia, 06123 Perugia, Italy; maria.pacilli@unipg.it; 3Department of Neurosciences, Imaging and Clinical Sciences, University of Chieti-Pescara, 66100 Chieti, Italy; antonio.aquino@unich.it

**Keywords:** COVID-19, SARS-CoV-2, coronavirus, environmental behavior, media influence, attitudes, emotions, fear

## Abstract

The COVID-19 outbreak has dramatically changed our life. Despite the rapid growth of scientific publications about medical aspects of the pandemic, less has been explored about the effects of media communication regarding COVID-19 on healthy behaviors. Yet, the scientific literature has widely debated on how media can influence people’s health-related evaluations, emotions, and behaviors. To fill this gap, the aim of this study was to investigate the relationships between media exposure, people’s attitudes and emotions toward media contents, and healthy behaviors related to the use of public spaces, such as avoiding crowded places, wearing face masks, and maintaining social distance. A questionnaire referring to these variables was administered to an opportunistic sample of 174 participants in Italy during the off-peak period of the COVID-19 outbreak and before restrictions to mobility were extended to the whole country. Results showed that media exposure, the perception of social initiatives of prevention, and moderate levels of fear increase healthier behaviors in the use of public spaces. Perceiving alarming information did not significantly predict healthy behaviors in the use of public spaces. Results are discussed with reference to the previous literature. Suggestions to media communication to increase preventive behaviors during emergencies are also provided.

## 1. Introduction

In this paper, a study on the effects of TV news and discussions about COVID-19 in Italy, in terms of cognitive, affective, and behavioral reaction of the public, is presented. Particular attention has been devoted to the understanding of psychological mechanisms of influence that can promote preventive healthy behaviors in the use of public spaces during an off-peak period of the outbreak (e.g., wearing a medical mask and keeping social distance). To this aim, the paper has been developed paying attention to different aspects discussed in the following sections: (1) the emergence of COVID-19, its diffusion, and negative consequences; (2) the mechanism of infection and then the central role of different uses of public spaces; (3) the role of the media in affecting the human response to COVID-19; (4) the psychological mechanisms of media influence that may play a role with reference to COVID-19; and (5) the research, with methods, results, discussions, and conclusion.

### 1.1. Spread and Consequences of COVID-19

SARS-CoV-2 infection (COVID-19), starting as an emergence in Wuhan, China, in December 2019 [1], has dramatically impacted human beings. It has rapidly and widely spread worldwide, causing global harm [2]. Fever, cough, shortness of breath, taste and smell dysfunction, and, in severe cases, bilateral pneumonia are the most common symptoms [3,4]. Overall, the World Health Organization (WHO) has reported more than 65 million cases and 1.5 million deaths worldwide [5].

Its impact has been huge on health-care systems, the economy, and personal lifestyles. Among the consequences, the health-care systems have suffered with regard to a delay in surgical care [6], the well-being of professionals [7,8,9], and the availability of medical devices [10]; the economic implications through the effects of the pandemic on restrictions for workers in agriculture, energy, manufacturing, education, hospitality, and tourism have been impressive [11]; the everyday lives of people have heavily changed across countries due to mobility restrictions [12,13]; and the impact on people’s mental health and well-being has also been extensively reported [14,15,16,17].

Among the mechanisms of infection, the role of contact with infected droplets during social interactions is of central importance [18,19]. In this regard, behavioral changes in the use of public spaces have attracted paramount attention in public information and initiatives to control the pandemic, as they can contribute to a reduction in social contacts and droplet exchange. Restrictions on human mobility, including travel limitations and social isolation, have been found to affect significantly the control of COVID-19 spread [20]; ventilation in indoor public spaces, such as work environments, is positively associated with a decrease in transmission [21,22]; the distance between tables and ventilation have been identified as key measures to prevent infection in public places such as restaurants [23]; and wearing a medical mask and keeping social distance have been widely claimed as the most effective control measures in public open spaces [24,25]. Overall, the role of urban crowding in the spread of the pandemic has been largely discussed [26,27].

### 1.2. COVID-19 and the Role of Media Influence

Despite the claim of a multidisciplinary approach to human health and well-being in facing the spread of COVID-19, little is known about the potential psychological effects of exposure to media news and discussions about the pandemic. Yet, their effect on (un)healthy uses of public spaces related to the COVID-19 pandemic may be huge. In their influential paper, Holmes et al. [28] clearly identified several immediate priorities for mental health research referring to COVID-19. In regard to the role of the media, they stressed the need to “understand the role of repeated media consumption in amplifying distress and anxiety, and optimal patterns of consumption for wellbeing; develop strategies to prevent over-exposure to anxiety-provoking media, including how to encourage diverse populations to stay informed by authoritative sources they trust; mitigate and manage the effect of viewing distressing footage” and to “understand how health messaging can optimize behavior change, and reduce unintended mental health issues; track perceptions of and responses to public health messages to allow iterative improvements, informed by mental health science” (p. 550). A similar concern has been expressed by other scholars [29,30]. So far, a handful of studies has followed these recommendations. Interestingly, most of the research has been devoted to the effects of misinformation and fake news about COVID-19 [31,32,33,34], instead of considering traditional media influence.

Regarding the research on traditional media influence, a study among Chinese respondents during the off-peak period of the COVID-19 outbreak outlined a positive association between TV exposure to news about COVID-19 and perceived susceptibility to the likelihood of getting infected, perceived severity of the disease, self-efficacy, and perceived control over the implementation of preventive measures. Moreover, TV exposure led to lower perceived emotional consequences associated with the infection [35]. Chao et al. [36] reported a mediational effect of media use on the relationship between the state of boredom and psychological distress among Chinese adults in the initial phase of the COVID-19 outbreak. Liu and Liu [37] found an effect of media exposure on anxiety through the vicarious traumatization mechanism, when the audience views reports and emotional visual materials about COVID-19 referring to other persons. Another study conducted in China one week after the declaration of person-to-person transmission of COVID-19 [38] showed a significant association of social media, but not traditional media, with negative outcomes. The nature of the content played a central role. Stressful content, such as severity of the outbreak or reports from hospitals, was associated with a more negative affect and depression. On the other hand, heroic acts, speeches from experts, and knowledge of the disease and prevention measures were associated with a more positive affect and less depression. Moreover, media engagement, expressed in terms of content sharing and search for news updates, was associated with more negative effects. In their study among the German population during nationwide restrictions, Bendau et al. [39] showed that the frequency, duration, and diversity of media exposure were positively associated with symptoms of depression and both unspecific and COVID-19-related anxiety. The authors also identified a critical threshold between mild and moderate symptoms, referring to seven times per day and 2.5 h of media exposure. Besides, exposure to social media was associated with higher psychological distress. Similarly, Ahmad and Murad [40] found a significant impact of social media on spreading fear and panic related to COVID-19, with a potential negative influence on mental health and psychological well-being among the young population of the Kurdistan region of Iraq. Croucher et al. [41] also found that social media use has a significant influence on prejudice toward Chinese Americans, who were blamed for carrying and spreading the virus.

### 1.3. Psychological Mechanisms of Media Influence and Healthy Behaviors: Towars a Comprehensive Framework for COVID-19

The role of the media in shaping psychological processes and human behavior has been extensively documented with reference to a variety of domains [42,43,44,45,46,47]. Classical theories have been proposed for decades in this regard. Cultivation theory argues that the media has massive, long-term, and cumulative effects, influencing a large and heterogeneous public through exposure to recurrent patterns of stories, images, and messages [48]. In this theory, light and heavy viewers are distinguished. For heavy viewers, the integrated content of television reaches all levels of our society and cultivates diverse social groups to share common views and conceptions of reality. Similarly, the agenda-setting theory postulates that the public learns not only about a given issue but also how much importance to attach to that issue from the amount of information in a news story and its position [49]. Health-related behaviors are not an exception, the effectiveness of campaigns about safe driving and alcohol consumption being appropriate examples [50,51]. The literature has identified several psychological mechanisms of media influence that can be highly relevant with reference to COVID-19-related healthy behaviors in the use of public spaces.

#### 1.3.1. Mere Exposure and Cognitive Overload

The outbreak of COVID-19 has led individuals to search for health information by increasing their health literacy, that is, the ability to access, review, and use health information [52]. However, the pandemic can be also associated with an infodemic, namely the dissemination of an enormous amount of information coming from different sources and whose foundation is often not verifiable. Just like viruses, news today spreads very quickly and through multiple channels. The information contagion has the effect of making emergency management much more complex, as it affects the possibility of transmitting clear and unambiguous instructions and thus obtaining homogeneous behavior on the part of the population [53,54]. Indeed, media coverage of COVID-19 news has been unprecedented across countries. Although the public needs information from trusted sources to make decisions about healthy behaviors to carry out, mere exposure to a significant amount of coverage may lead to overload, and much more when information is novel, inconsistent, or ambiguous [30]. Information overload has been defined as a “situation that arises when an individual’s efficiency and effectiveness in using information is hampered by the amount of relevant, and potentially useful, information available to them” [55] (p. 12). It has been investigated for decades in different disciplines, including psychology, organizational science, and economics [56]. The negative consequences of information overload have been extensively outlined and pertain to emotional states, cognitive activity, and behavior; among the others, anxiety, mental fatigue, impairment of attention and reasoning, inefficiency, sharing of misinformation, and irrational behavior have been documented [55,56,57].

#### 1.3.2. Cognitive Evaluations of TV News and Discussions about COVID-19.

Media exposure can shape the social perception of an event, expressed in terms of judgments or attitudes that, in turn, affect behavior [58,59,60]. Attitudes have been defined in social psychology as “a psychological tendency that is expressed by evaluating a particular entity with some degree of favor or disfavor” [61] (p. 1). In his insightful contribution to the effects of media exposure, Shrum [62] convincingly discussed how heuristics and accessibility are basic processes of social cognition; namely, when people develop judgments, they typically rely on a small subset of information available, and the frequency [63], recency [64], and vividness of information through extreme examples [65] make them more readily accessible to mind. When attitudes are formed and made accessible, they can affect behavior. This applies also to TV news and discussions about COVID-19.

#### 1.3.3. Emotions Elicited by News and Discussions about COVID-19

Two very different emotions may play a key role in the audience’s reactions. One emotion is fear, and the above literature has started investigating its role and consequences with reference to news about COVID-19. Overall, the influence of fear spread by the media on behavior has been largely analyzed with reference to fear appeals. Fear appeals can be defined as persuasive communications that refer to a dangerous practice, depict its negative consequences, and describe how those outcomes can be avoided through proper behavior [66]. Fear depends on perceived threat and perceived efficacy. The former is composed of two dimensions: perceived susceptibility to the threat, namely the degree to which one feels at risk, and the perceived severity of the threat, referring to the magnitude of harm expected from the threat. Perceived efficacy is composed of two dimensions as well: perceived self-efficacy, referring to one’s beliefs about the ability to perform the proper behavior, and perceived response efficacy, namely one’s beliefs about whether that behavior can be effective in avoiding the threat [67]. The seminal study on fear appeals has shown that lower levels of fear are more effective than those evoking a high fear response, in terms of both immediate attitude change and resistance to subsequent counter-attitudinal messages [68]. These, and subsequent, results have been predominantly explained through a fear drive model. According to this model, persuasion increases as fear increases from low to moderate levels, but beyond a specific threshold, the strong negative emotional response can be more easily controlled by discounting, rather than following, the message [69]. Subsequent research was inconsistent, with fear and persuasion positively associated, and the issue of optimal fear levels, or thresholds, has been widely investigated, and the role of moderators, as the credibility of the source or self-esteem of recipients, as well [66,70,71]. For example, among cognitive processes, the individual perception of collective efficacy has been found to interact with higher fear levels in predicting the higher intention to engage in pro-environmental behavior among Chinese respondents [72]; with reference to emotions, disgust has emerged to increase persuasion in fear appeal messages [73]. On the whole, recent meta-analytic reviews of the literature have stressed the importance of effectiveness of recommended behaviors and perceived self-efficacy as motivational forces to promote a behavioral change; namely, people must know what to do, think they can, and believe it is effective [74,75]. Another relevant issue at stake is trust in authorities. Recent research [76] has shown, indeed, that during the 2014–2015 Ebola epidemic, trust in Liberian authorities was a crucial predictor of citizens’ compliance with policies for containing the disease.

### 1.4. Research Aims and Hypotheses

Overall, media coverage of COVID-19 may have an effect on the public through different processes that should be investigated within a comprehensive framework. Therefore, the aim of this study was to investigate the relationships between exposure to TV news and discussions about the COVID-19 pandemic in Italy, the way in which media content has been evaluated by the audience, their emotional reactions in terms of fear or trust, and the overall effects on people’s use of public spaces as preventive behaviors to control the infection. Because media content can rapidly change, our aim was to better understand these relationships within specific conditions, namely at the start of the outbreak and for a population somewhat distant from the pandemic.

From the literature, it is possible to hypothesize that behavioral responses in terms of the use of public spaces to prevent the infection are affected by mere exposure to news and discussions about the pandemic, the audience’s attitudes toward the content, and the emotions evoked, which operate at a different level as distinct yet integrated psychological processes.

## 2. Materials and Methods

### 2.1. Study Design

A cross-sectional study using an opportunistic sample method was conducted between February and March 2020 in Rome, Italy.

### 2.2. Participants

Participants were recruited among relatives and friends of students attending a course of social psychology in Rome. No reward was given for participation. Overall, 576 questionnaires were given to students. Students were asked to distribute the questionnaires to relatives and friends of different sexes and ages and to collect them within a couple of days. Time was a relevant aspect, because the media content about COVID-19 was supposed to change rapidly. An opportunistic sample of 174 respondents returned the questionnaire (62.6% females, age range 17–86 years old, mean age 37.4, SD 16.4, educational level: 42.5% high school or lower, 27% bachelor’s degree, 30.5% master’s degree).

### 2.3. Procedure

Participants were asked to fill in a paper-and-pencil questionnaire on attitudes toward TV news and discussions about the COVID-19 pandemic in Italy and their reaction in terms of emotions and preventive behavior. No questionnaire was found to be incomplete and thus excluded from the analyses. The data collection was completed by 4 March 2020, before restrictions to mobility were extended to the whole country (Figure 1).

### 2.4. Tools

A questionnaire for the measurement of exposure to TV news and discussions about the COVID-19 pandemic, their evaluations, emotional reactions, and behavioral changes from the public was developed. With reference to media content, we focused on recurrent themes in international and national media coverage about the virus’s origins, trajectory, and impact [78]. As a consequence, we selected three macro-themes discussed in TV news in Italy from February 20 to 26, 2020, when the COVID-19 pandemic had spread in several zones of northern Italy, but no national anti-contagion policies had been proposed yet. They referred to the contagion in itself in the quarantine (called red) zones and its immediate consequences (e.g., the number of people infected, the lockdown, the strong decrease in the use of public transport), the preventive measures suggested to people living in COVID-19-free (called green) zones (e.g., the importance of washing one’s hands, wearing a face mask, and keeping social distance), and the economic effects of the COVID-19 pandemic (e.g., the stock market crash, the negative effects on tourism and work activities). Several items for each macro-theme, referring to attitudes of people toward the topic addressed, were developed; items referring to emotional reactions and behavioral changes to the COVID-19 pandemic, in terms of the use of public spaces, were developed as well (see Section 2.2).

The above-mentioned items were organized in different sections of the questionnaire distributed among participants. Section 1 included 47 items that mapped participants’ attitudes toward the three macro-themes of TV news and discussions about the COVID-19 pandemic in Italy. Items were introduced by the sentence “In the days following the first cases of infection from COVID-19 in Northern Italy, TV news and discussions have stressed…” With reference to the contagion in the red zones, example items are “The number of infected people” and “The number of swab-tested people”; with reference to the preventive measures suggested to people living in green zones, example items are “The importance of washing your hands” and “The importance of keeping social distance”; with reference to the economic effects of the COVID-19 pandemic, example items are “The stock market crash as a result of coronavirus” and “The negative effect of coronavirus on work activities”.

Section 2 included 4 items measuring the fear of COVID-19 (an example item is “I am afraid that coronavirus may spread largely in my city”), 4 items measuring trust in the authorities and their policies to control the diffusion of COVID-19 (an example item is “The Italian government has the competence to control the coronavirus emergency”), 4 items measuring behavioral changes in the use of public spaces to prevent infection (an example item is “I go to restaurants or public places less often than before”), and a single item measuring media exposure, namely the amount of time devoted to watching TV news and discussions about the COVID-19 pandemic in Italy.

Section 3 included socio-demographics (gender, age, educational level, work activity, and place of residence).

All attitude, emotional, and behavioral items included in Section 1 and Section 2 were rated on a 7-point scale ranging from 0 = Completely disagree to 7 = Completely agree.

### 2.5. Statistical Analyses

Several statistical analyses were conducted for the aims of the study. Exploratory factor analyses (EFAs) were performed to check the dimensionality of attitudes toward TV news and discussions about the COVID-19 pandemic in Italy. Separate EFAs were performed referring to the three macro-themes identified. Separate EFAs were performed with reference to emotional and behavioral reactions of the audience as well.

Reliability analyses (RAs) through the scoring of Cronbach’s alpha were performed to check the internal consistency of the dimensions that emerged from EFAs. Mean scores of all the dimensions were then computed.

Finally, correlational analyses (CAs) and a final hierarchical multiple regression analysis (HMRA) were performed to investigate the overall pattern of relationships between the audience’s exposure to TV news and discussions about the COVID-19 pandemic in Italy, their attitudes toward the way in which they have been presented, their emotional consequences in terms of fear or trust, and their effects on people’s use of public spaces as preventive behaviors to control the infection.

### 2.6. Ethical Aspects

The study was conducted in accordance with the ethical standards of the 1964 Declaration of Helsinki, and it fulfilled the ethical standard procedure recommended by the Italian Association of Psychology (AIP). Before taking part in the study, participants were informed of their right to refuse to participate in the study or to withdraw consent to participate at any time during the study without reprisal.

## 3. Results

### 3.1. Factor and Reliability Analyses

Three EFAs were performed on items measuring attitudes toward TV news and discussions about the COVID-19 pandemic in Italy. The three EFAs referred to the macro-themes Contagion in Red Zones and Its Consequences, Preventive Measures in Green Zones, and Economic Effects of COVID-19.

With reference to the Contagion in Red Zones and Its Consequences macro-theme, 22 items were included in the EFA with Varimax rotation. The Kaiser-Meyer-Olkin (KMO) measure of sampling adequacy (0.83) and Bartlett’s test of sphericity (approx. chi-square = 1234.7; df = 171) were both significant (*p* < 0.001). Both the scree plot and factor loadings suggested a two-factor solution, explaining 43.9% of the variance (Appendix A). The two dimensions were labeled (1) *calming information*, stressing the positive aspects of the contagion (e.g., even in red zones, the majority of residents had no symptoms and deceased persons already suffered from other pathologies), and (2) *alarming information*, referring to the danger of COVID-19 (e.g., the increase in deceased persons, the ease of diffusion, the limitations of movement). The dimensions were not significantly correlated. Two items were excluded from the final solution, namely the *contagion of healthcare workers* and the *absence of immune defense against COVID-19*. RAs showed a good level of internal consistency for both dimensions emerging from the EFA.

With reference to the Preventive Measures in Green Zones macro-theme, 16 items were included in the EFA with Oblimin rotation. The KMO measure of sampling adequacy (0.81) and Bartlett’s test of sphericity (approx. chi-square = 973.5; df = 120) were both significant (*p* < 0.001). Both the scree plot and factor loadings suggested a three-factor solution, explaining 55.4% of the variance (Appendix A). The three dimensions were labeled (1) *individual prevention*, stressing the importance of respecting several health-related measures to prevent infection, such as washing one’s hands, avoiding touching one’s mouth and nose, and keeping social distance; (2) *social prevention*, referring to measures implemented by the authorities in public spaces, such as the sanitization of public transport and the control of temperature in public spaces; and (3) *institutional restrictions*, addressing government interventions in terms of closing of public exercises and cultural initiatives, such as restaurants, museums, and schools. The dimensions showed significant correlations. No item was excluded from the final solution. RAs showed a good level of internal consistency for all the dimensions emerging from the EFA.

With reference to the Economic Effects of COVID-19 macro-theme, 8 items were included in the EFA with Oblimin rotation. The KMO measure of sampling adequacy (0.78) and Bartlett’s test of sphericity (approx. chi-square = 619.1; df = 28) were both significant (*p* < 0.001). Both the scree plot and factor loadings suggested a two-factor solution, explaining 64.0% of the variance (Appendix A). The two dimensions were labeled (1) *economic effects*, stressing the impact of the COVID-19 pandemic on working activities, tourism, and the stock market, and (2) *price increase*, referring to the increasing cost of medical devices such as face masks and sanitizing gel. The two dimensions showed a significant correlation (*r* = 0.31). No item was excluded from the final solution. RAs showed a good level of internal consistency for the two dimensions emerging from the EFA.

Another EFA with Varimax rotation was performed on items measuring relevant emotions raised by TV news and discussions about the COVID-19 pandemic in Italy, with reference to fear and trust in authorities. With reference to emotions, 8 items were included in the EFA. The KMO measure of sampling adequacy (0.77) and Bartlett’s test of sphericity (approx. chi-square = 849.6; df = 2) were both significant (*p* < 0.001). Both the scree plot and factor loadings suggested a four-factor solution, explaining 64.6% of the variance (Appendix A). The two dimensions were labeled (1) *fear*, stressing the fear of an increased diffusion of COVID-19 across cities and among friends and relatives, and (2) *trust in authorities*, referring to the perceived capability of the government to adequately manage the pandemic at local, national, and EU levels. The two dimensions did not show significant correlation. No item was excluded from the final solution. RAs showed a good level of internal consistency for the two dimensions emerging from the EFA.

A final EFA was performed on items measuring behavioral change in the use of public spaces, with 8 items included in the analysis. The KMO measure of sampling adequacy (0.89) and Bartlett’s test of sphericity (approx. chi-square = 878.5; df = 28) were both significant (*p* < 0.001). Both the scree plot and factor loadings suggested a one-factor solution, explaining 61.4% of the variance (Appendix A). The dimension was labeled *change in the use of public spaces*, stressing the behavioral change in everyday habits when using public spaces to prevent infection, such as going to restaurants or cinemas or using public transport or open spaces. No item was excluded from the final solution. RA showed a good level of the dimension emerging from the EFA.

### 3.2. Correlational Analysis (CA) and Hierarchical Multiple Regression Analysis (HMRA)

To investigate the influence of TV exposure to news and discussions about COVID-19 (*M* = 104 min per day, SD = 93 min) on the change in the people’s use of public spaces as preventive behaviors against the infection, and considering the role of attitudes and emotions promoted by TV communication, correlational analysis (CA) and hierarchical multiple regression analysis (HMRA) were performed.

The correlation matrix showed several significant correlations between the variables considered (Table 1). TV exposure was positively correlated with institutional restrictions (r = 0.16, *p* < 0.05), fear (r = 0.20, *p* < 0.05), and change in the use of public spaces (r = 0.37, *p* < 0.01). Several significant associations were found among the dimensions of attitude, with the highest correlation emerging between alarming information and institutional restrictions (r = 0.52, *p* < 0.01). Among the different dimensions, calming information was significantly correlated with fear (r = 0.29, *p* < 0.01) and change in the use of public spaces (r = 0.17, *p* < 0.05); individual prevention was significantly correlated with trust in authorities (r = 0.21, *p* < 0.01), fear (r = 0.24, *p* < 0.01), and change in the use of public spaces (r = 0.16, *p* < 0.05); social prevention was significantly correlated with trust in authorities (r = 0.18, *p* < 0.05), fear (r = 0.16, *p* < 0.05), and change in the use of public spaces (r = 0.24, *p* < 0.01); and economic effects was significantly correlated with fear (r = 0.16, *p* < 0.05). Neither alarming information nor price increase showed a significant correlation with emotions and change in the use of public spaces.

Finally, hierarchical multiple regression analysis (HMRA) was performed to assess the overall relationships between TV exposure to news and discussions about COVID-19 (step 1), attitudes (step 2), emotions (step 3), and behavioral changes in the use of public spaces (Table 2). The variables that did not show a significant association with the criterion were not included in the HMRA. At step 1, the model was significant and TV exposure emerged as a significant predictor of activism. At step 2, the model significantly increased the amount of explained variance, with social prevention emerging as a significant predictor of activism. TV exposure was still significant. At step 3, the model still increased the amount of explained variance, and fear emerged as a significant predictor of change in the use of public spaces. TV exposure and social prevention still remained significant predictors of change in the use of public spaces.

## 4. Discussion

This study helped shed light on the potential effects of media exposure to news and discussions about COVID-19 on people’s healthy behaviors related to the use of public spaces, through data collection during the off-peak period of the outbreak in Italy, when the pandemic had started spreading only in northern Italy. Media influence has been widely investigated with reference to classical theories of communication, postulating a variety of effects on the public [48,49]. The previous literature stressed the role of several psychological mechanisms of media influence, referring to mere exposure [30,55,57] and both cognitive evaluation of [62,63,64,65] and emotions perceived in association with [72,73,74,75] media content. As a consequence, in this study, the news was analyzed in terms of themes discussed on TV programs; people’s reactions to exposure were investigated in terms of attitudes and emotions toward the content of TV news and discussions and their effects on preventive behaviors in public spaces to control the infection.

EFAs were preliminarily performed to identify relevant dimensions of attitudes and emotions toward TV news and discussions about COVID-19 and behaviors to prevent infection; several items referring to attitudes, emotions, and behaviors have been developed to this end.

Three EFAs were carried out with reference to attitudes, referring to themes largely discussed in TV news in Italy. They pertained to the contagion in red zones and its immediate consequences (e.g., the number of people infected, the lockdown, the strong decrease in the use of public transport), the preventive measures for people living in green zones (e.g., the importance of washing one’s hands, wearing a face mask, and keeping social distance), and the economic effects of the COVID-19 pandemic (e.g., the stock market crash, the negative effects on tourism and work activities). Overall, the dimensions of attitude emerging from EFAs reflected several aspects of media communication widely considered by previous research.

With reference to the contagion in red zones and its consequences, two dimensions of attitude emerged, showing a good level of internal consistency. On the one hand was calming information, emphasizing, for example, that even in red zones, the majority of residents had no symptoms or deceased persons already suffered from other pathologies. On the other hand was alarming information, where the danger of COVID-19, expressed in terms of the ease of diffusion, the increase in deceased persons, and the importance of personal restrictions, was dramatically outlined. While information resting on alerts and emotionality has been extensively reported in the literature on emergencies, the role of calming information has remained largely unexplored. Negative effects of news emotionality on the audience have been identified with reference to bioterrorism [79], bacterial infections [80], and terrorist attacks [81,82,83]. For example, both Cho et al. [81] and Gadarian [82] have reported higher levels of negative emotions such as fear and anger among respondents exposed to emotional news about the 9/11 terrorist attack in the U.S. Taken together, these dimensions are compatible with positive vs. negative framing of information, whose effects have been largely investigated in research on communication with reference to a variety of issues [84,85,86,87,88].

With reference to preventive measures in green zones, three dimensions of attitude emerged, also showing a good level of internal consistency: individual prevention, based on several health-related measures against the infection, such as washing one’s hands, avoiding touching one’s mouth and nose, and keeping social distance; social prevention, referring to measures implemented by the authorities in public spaces, such as the sanitization of public transport and the control of temperature in public spaces; and institutional restrictions, addressing political interventions to close public exercises and stop cultural initiatives. This distinction can be highly relevant as the interplay between interventions at different levels against infections has shown to be effective in the past—to give just an example, with reference to HIV [89,90,91].

With reference to the economic effects of COVID-19, two dimensions of attitude emerged, showing a good level of internal consistency: economic effects, referring to the impact of the COVID-19 pandemic on the stock market, work activities, and tourism, and price increase, stressing the increasing cost of medical devices to be used to face the infection, such as face masks and sanitizing gel. Past research on the media coverage of emergencies has shown potentially dramatic effects of this communication on the economy, as in the case of terrorism [92,93]. For example, Melnick and Eldor [92] stressed that rather than terrorist attacks in themselves, it is the media coverage of terrorism that is the only significant variable explaining the economic damage incurred.

Two final EFAs were carried out for emotions and behaviors, respectively. The former yielded two dimensions, referring to trust in authorities and fear of COVID-19. Both confirmed psychological processes widely investigated in the literature on media effects [67,70,71,72,74,75,76]. The latter identified a single dimension of preventive behaviors toward COVID-19, including avoiding crowded places, using face masks in public spaces, and keeping social distance, in addition to a difference in the use of public transport and working habits.

CA and HMRA have suggested a comprehensive image of how these psychological processes may affect healthy behaviors related to the use of public spaces. CA has shown a significant association between media exposure, dimensions of attitude toward TV content, emotions elicited, and healthy behaviors. Among the dimensions of attitude, calming information, individual prevention, and social prevention were positively correlated to healthy behavior. Interestingly, giving alarming information has no increasing effect on healthy behaviors. This result is compatible with previous research showing that emotional communication of emergencies can foster negative effects [78,79,80,81,82]. The final HMRA identified the relevance of the three basic mechanisms investigated in this study, namely mere exposure, evaluations of and emotions toward TV news, and discussions about COVID-19, for healthy behaviors related to the use of public spaces. In particular, exposure, social prevention, and fear emerged as significant predictors of healthy behaviors in the final HMRA. The role of exposure has been systematically addressed by previous research, also with reference to news about COVID-19. Our result is compatible with the study of Bendau et al. [39], indicating that time exposure above a relevant threshold can promote negative effects; below this threshold, as in the case of the present study, mechanisms of information overload and negative emotions are less likely to occur, and exposure may turn into positive behaviors. In general, the positive role of mild exposure during an off-peak period of the outbreak is in line with [35], while in full emergency, the influence may become negative [37,38]. Social prevention referred to the idea that the overall community is facing together the COVID-19 outbreak by the implementation of initiatives where individual and social health are inextricably intertwined. This evaluation in the audience increased heathy behaviors in public spaces. This result is compatible with research based on social identity theory [94] that shows that when people perceive a stronger identification with others, prosocial behavior increases [95,96]; and a change in the use of public spaces during the COVID-19 outbreak can be considered a kind of prosocial behavior. More recently, a significant relationship between social identification and emotional well-being was outlined with reference to COVID-19 [97]. The role of calming information and individual prevention, which was significant in CA, did not emerge in the final HMRA. This can be due also to statistical reasons, where multiple predictors concur. Overall, the importance of prevention at different levels, also stressed in previous research [89,90], needs further exploration with reference to COVID-19. Finally, as fear increased, healthy behaviors in public spaces increased. This result suggested that with adequate levels of information and positive evaluations of content, fear can have a strong adaptive value [98,99,100].

This study has also several limitations that should be highlighted. First, the use of convenience sampling with a high proportion of young participants in this study may introduce a bias. Second, the study was based on self-reporting, which may be subject to bias as well. Third, the associations found in this study should be interpreted with caution as the responses were obtained during the off-peak period of the COVID-19 outbreak. It is possible to hypothesize very different patterns of associations between exposure, evaluations, emotions, and preventive behaviors in the use of public spaces during periods of widespread COVID-19. In addition, media coverage changed rapidly in themes and modes of presentation as the pandemic went on. Fifth, we investigated the influence of TV news and discussions, but the role of other sources of information (e.g., social media) is probably huge. Future research will have to better understand all these issues. Despite these limitations, this study contributed tremendously to the understanding of the influence of mass media on responses to the COVID-19 outbreak.

## 5. Conclusions

This study contributed to identifying some key psychological mechanisms of media influence on healthy uses of public spaces, with reference to TV news and discussions about COVID-19. Mere exposure, positive attitudes toward social prevention promoted by TV content, and moderate levels of fear emerged as the variables most associated with positive behaviors in public spaces to prevent the risk of contagion. Calming information and indications for individual prevention also showed a significant association with healthy behaviors in correlation analysis, while alarming information was ineffective.

Taken together, these results can have relevant potential applications in the management of communication during an initial phase of a global emergency, as in the case of the COVID-19 outbreak. Optimal amounts of media exposure should be recommended to the audience. News content, and its emotional tone as well, should be carefully considered by the media when informing people about an emergency. Giving the idea that the community can better face an emergency if it works like an organism, where every single component plays its role, can be important as well. This can have a strong impact on preventive behaviors during off-peak periods of an emergency, when a positive collective reaction of people is fundamental to limiting harmful outcomes. The consequences on the population, in terms of healthy vs. unhealthy influences, may be huge.

## Figures and Tables

**Figure 1 ijerph-18-01879-f001:**
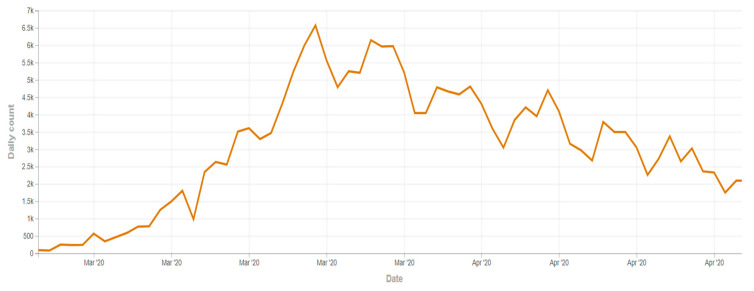
Trend of reported infections of COVID-19 in Italy and data collection of the study (adapted from [77]).

**Table 1 ijerph-18-01879-t001:** Correlation analysis: TV exposure, people’s attitudes, emotions, and the use of public spaces.

	1	2	3	4	5	6	7	8	9	10	11
1. TV exposure	1										
2. Alarming information	0.07	1									
3. Calming information	0.10	−0.01	1								
4. Individual prevention	0.13	0.40 **	0.30 **	1							
5. Social prevention	0.03	0.24 **	0.40 **	0.40 **	1						
6. Institutional restrictions	0.16 *	0.52 **	0.22 **	0.38 **	0.38 **	1					
7. Economic effects	0.01	0.44 **	0.20 *	0.36 **	0.27 **	0.40 **	1				
8. Price increase	0.19	0.39 **	0.08	0.13	−0.08	0.21 **	0.41 **	1			
9. Trust in authorities	−0.09	0.05	0.07	0.21 **	0.18 *	0.12	−0.01	−0.07	1		
10. Fear	0.20 *	0.14	0.29 **	0.24 **	0.16 *	0.09	0.16 *	0.10	0.05	1	
11. Change in the use of public spaces	0.37 **	0.10	0.17 *	0.16 *	0.24 **	0.06	0.14	0.03	−0.01	0.48 **	1

*: *p* < 0.05; **: *p* < 0.01.

**Table 2 ijerph-18-01879-t002:** Hierarchical multiple regression analysis: predictors of change in the use of public spaces.

Predictors	*β*-Coefficients			Adjusted R^2^	R^2^ Change
	Step 1	Step 2	Step 3		
*Step 1*				0.13 ***	
TV exposure	0.37 ***	0.36 ***	0.29 ***		
Step 2				0.17 ***	0.04 *
Calming information		0.05	−0.05		
Individual prevention		0.02	−0.04		
Social prevention		0.21 *	0.20 **		
Step 3				0.32 ***	0.15 ***
Fear			0.41 ***		

***: *p* < 0.001; **: *p* < 0.01; *: *p* < 0.05.

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
