# Peer review of "TV News and COVID-19: Media Influence on Healthy Behavior in Public Spaces"

_ijerph, 2021, doi:10.3390/ijerph18041879_

Round 1
Reviewer 1 Report
This manuscript results from the construction, usage and analysis of a questionnaire on the perceived media influence on healthy behavior on public spaces. It consists of three parts. In the first part, an overview is given of selected publications on the way COVID-19 influences society. In the second part the authors report on their own interview with students, friends and other relations. In the third part the reported data are analyzed and some conclusions are tentatively formulated.
The manuscript fails in paving a scientific foundation for the paper. It claims facts that are debatable (for instance ref [1] poses that the virus comes from China while earlier occurrences are noted). The literature search is sometimes a ‘laudatio’ instead of fact finding. Above all, the paper misses a clear problem statement, leading eventually a verification of available conclusions. Actually, the discussion of experimental results are presented after the conclusions.
The authors provide a long list of published papers in the area. This is meaningful for the view readers that do not have search machines available. For the average reader, most of the given references make no sense as they are only ‘called’ in the manuscript but are not linked to provide a context for the discourse. Further each sample is sociologically colored and is therefore representative, rather indicative.
The questionnaire samples only a single moment in time. However, the world in which COVID-19 is conceived changes continuous. This has two consequences. Firstly, some of the references give conclusions that reflective the emotions at a specific moment in time. Arbitrarily combining such references runs the risk of confusing issues. Secondly, the facts derived from the current questionnaire are part of the past and may easily have become invalid by socio-cultural movements.
The following is a non-exhaustive list of language errors:
[Section 1.2; 2nd line] “In this regards” è “in this regard”
[Page 2; 6st line from bottom] It is not clear what is mild and what is moderate.
[Page 6; line 7 from bottom] “two item” è “two items”
[Page 8; line 11 after table 3] internal consistency is caused by temporal social coherence. Are the found correlations really meaningful?
[Section 4] “helped shed light” è “helped to shed light”
[Page 11; line 7 from bottom] This is really hard to understand. I am not sure whether it is just punctuation or even word order.
[Page 11; line 6 from bottom] “Showed”?
[Page 12] If causality is missing, how about using some simple and straightforward plausibility?
[Section 5; line 2] ?Mere?
In general, there are a number of shortcomings with the research design. Moreover, the manuscript is not written in a way that makes for easy understanding. Sentences are too long. Discussion does not lead to conclusions while the conclusion is a discussion about the research field in general. I was told by a specialist that the paper has in its writing not the level generally expected in this niche of medical sciences.
Author Response
Reviewer 1
This manuscript results from the construction, usage and analysis of a questionnaire on the perceived media influence on healthy behavior on public spaces. It consists of three parts. In the first part, an overview is given of selected publications on the way COVID-19 influences society. In the second part the authors report on their own interview with students, friends and other relations. In the third part the reported data are analyzed and some conclusions are tentatively formulated.
The manuscript fails in paving a scientific foundation for the paper. It claims facts that are debatable (for instance ref [1] poses that the virus comes from China while earlier occurrences are noted).
Scientific evidence about the first cases is continuously updating. Recently, infections in Italy in November 2020 were also reported. Probably, it will be hard to say for sure – and once and for all - where the first case comes from. We are aware of this, and then we avoided stating something in this regard. Our sentence carefully stressed where COVID-19 started "as an emergence", that is Wuhan, China. This is undoubtedly a fact.
The literature search is sometimes a ‘laudatio’ instead of fact finding. Above all, the paper misses a clear problem statement, leading eventually a verification of available conclusions. Actually, the discussion of experimental results are presented after the conclusions. The authors provide a long list of published papers in the area. This is meaningful for the view readers that do not have search machines available. For the average reader, most of the given references make no sense as they are only ‘called’ in the manuscript but are not linked to provide a context for the discourse.
Thank you for this request of clarification. We have now tried to better explain the sequence of arguments of the paper. In this regard, we added a short paragraph at the beginning of the introduction (rows 30-39), to give an overall context of the discourse.
We carefully selected the literature following the aim of the paper, and according to a sequence of arguments, namely:
- The emergence of COVID-19, its diffusion and huge consequences.
- The mechanism of infection, and then the central role of different uses of public spaces (our dependent variable).
- The role of media influence in affecting human response to COVID-19
- The psychological mechanisms of influence investigated by the literature (that we applied in this study on media influence with reference to COVID-19).
The problem statement was better indicated (rows 30-34).
Further each sample is sociologically colored and is therefore representative, rather indicative.
We are aware that the sample is opportunistic and we highlighted this aspect in the limitations of the study. However, as discussed also in the following point, we were interested in understanding people response to communication about COVID-19 at a particular phase of the outbreak, in order to understand the role of psychological processes in promoting healthy behaviors in the use of public spaces (see new Section 1.4). As communication has rapidly changed, we needed to collect in a very short time as many respondents as possible to understand these relationships, even though the sample was not representative. However, we added more information about the sample, to show some variability (rows 231-233).
The questionnaire samples only a single moment in time. However, the world in which COVID-19 is conceived changes continuous. This has two consequences. Firstly, some of the references give conclusions that reflective the emotions at a specific moment in time. Arbitrarily combining such references runs the risk of confusing issues. Secondly, the facts derived from the current questionnaire are part of the past and may easily have become invalid by socio-cultural movements.
As the Reviewers have stressed, the world of COVID-19 has changed continuously. We agree. That's why our aim was to give a picture of this world under clear conditions: at the start of the outbreak, for a population somewhat "distant" from the pandemic, and when communication was about specific themes. This choice was made in order to understand the psychological processes that could better link TV news, people's reactions, and healthy behaviors. Understanding prevention was the main objective of our investigation. We have tried to better discuss these points in the new subsection Research aims and hypotheses (rows 188-200), as asked by Reviewer 2. Nonetheless, we are aware that the picture is not a movie, and thus we clearly commented on these aspects in the limitations of the study. The consequence of taking a picture under clear conditions is that we did not have much time. That's why we opted for a convenience sample. We recognized also this limitation in the discussion.
Reviewer 1 is right when saying that the references cited in the study refer to different moments of the pandemic. And different populations as well. We included all of them to give an overall picture of potential reactions at different moments and conditions. People's reactions are likely to be very different at the beginning of the pandemic vs. when it has largely spread in one's country, with several limitations to everyday life. From this starting point, our results can be better understood. In fact, they are compatible with studies conducted in similar conditions (off-peak periods), where negative emotional consequences were lower, self-efficacy and perceived control were higher (ref. 35) and other studies showing the relevance of a critical threshold of exposure between more positive vs. more negative outcomes (ref.39).
In the revised version of the paper, we have now tried to better link our findings to previous studies conducted in similar vs. dissimilar conditions (rows 462-464).
It is also true that our findings are valid in the conditions we chose to investigate. However, we do not think that they only belong to the past. As we pointed out in the revision of the discussion (rows 499-504), our findings can be useful also for the future, because in the event of another emergency, we identified mechanisms of response to news promoting healthy behaviors.
Unfortunately, it is less clear to us what the reviewer means by saying " invalid by socio-cultural movements ".
The following is a non-exhaustive list of language errors:
Thank you for these detailed indications. The above errors were amended and the language checked throughout the paper.
[Section 1.2; 2nd line] “In this regards” è “in this regard”
[Page 2; 6st line from bottom] It is not clear what is mild and what is moderate.
We used the same terms of the authors to indicate different levels of anxiety and depression.
[Page 6; line 7 from bottom] “two item” è “two items”
[Page 8; line 11 after table 3] internal consistency is caused by temporal social coherence. Are the found correlations really meaningful?
If Reviewer 1 suggested that all the measures were taken at the same time, that is just what we meant to measure.
[Section 4] “helped shed light” è “helped to shed light”.
Help shed light is correct. Help can be used with both the bare-infinitive and the to-infinitive.
[Page 11; line 7 from bottom] This is really hard to understand. I am not sure whether it is just punctuation or even word order.
Thank you for this suggestion. We hope that punctuation has made the sentence easier to understand, with reference to the three different themes.
[Page 11; line 6 from bottom] “Showed”?
[Page 12] If causality is missing, how about using some simple and straightforward plausibility?
[Section 5; line 2] ?Mere?
Mere exposure is a well-known process investigated in psychology.
In general, there are a number of shortcomings with the research design. Moreover, the manuscript is not written in a way that makes for easy understanding. Sentences are too long. Discussion does not lead to conclusions while the conclusion is a discussion about the research field in general. I was told by a specialist that the paper has in its writing not the level generally expected in this niche of medical sciences.
We tried to better clarify the research design and our choices in the methodology (rows188-196 and 224-234). We amended the language and split long sentences when possible (e.g. rows 82-87, 171-175, 210-215). We discussed the several results and the overall finding of the study (rows 493-499). We stressed in the conclusion the practical applications of these results (rows 499-504), while potential development of future research were addressed with reference to our limitations (rows479-490). We are not aware of the writing expected in medical sciences, because this study is psychological.
Reviewer 2 Report
Dear authors,
You did a good job. It would be interesting if you could include at the end of the introduction, a subsection with the research questions or the hypothesis you hope to refute with your work.
Example:
RQ1.....
RQ2....
RQ3.....
In order to enrich the introduction and stimulate the reading of the audience, it would be interesting if you could include some visual element such as a data visualization of the impact of the COVID-19 over time. Trying to contextualize it in your research with the media exposure would be a very attractive and powerful introduction!
My advice would be to try to take advantage of as many keywords as the IJERPH allows, as this will allow you to position your work in the various databases much better. For example, I would add: SARS-CoV-2, Coronavirus, emotions, sentiments...
In general the work is well documented, the results are well represented and the discussion is very interesting.
Best regards,
Author Response
You did a good job. It would be interesting if you could include at the end of the introduction, a subsection with the research questions or the hypothesis you hope to refute with your work.
Example:
RQ1.....
RQ2....
RQ3.....
Good suggestion to increase the readability of the paper. The subsection was added (rows 188-200).
In order to enrich the introduction and stimulate the reading of the audience, it would be interesting if you could include some visual element such as a data visualization of the impact of the COVID-19 over time. Trying to contextualize it in your research with the media exposure would be a very attractive and powerful introduction!
Thank you very much for this suggestion. A Figure showing the contagion increase over time in Italy has been added.
My advice would be to try to take advantage of as many keywords as the IJERPH allows, as this will allow you to position your work in the various databases much better. For example, I would add: SARS-CoV-2, Coronavirus, emotions, sentiments...
Thank you for this suggestion. A longer list of keywords was provided.
In general the work is well documented, the results are well represented and the discussion is very interesting.
Thank you very much for this comment!
Reviewer 3 Report
This is an interesting research topic. Some points need to be reviewed to improve the manuscript.
Abstract: It is not sufficiently clear. Please, introduction (01 line), and add aim, materials and methods and conclusion.
Introduction: Regarding the introduction, consider that it is well structured and organized. It allows the reader to find the information necessary to promote understanding of the state of affairs. As a suggestion, consider presenting short, precise paragraphs. What is currently known? What does this article add?
It is not sufficiently clear that they refer to the adult population.
Aim: It is not sufficiently clear in methods and abstract
Materials and Methods
I do, however, have some concerns/comments that I feel the authors need to address.
I believe that some more information on participants recruitment methods would be useful to the reader.
Was any kind of recompense for participation provided?
What are the inclusion and exclusion criteria? Who did the data collection?
Please include additional information regarding the survey or questionnaire used in the study and ensure that you have provided sufficient details that others could replicate the analyses. Who developed the questionnaire? Was there a pre-test?
I suggest Including that is a cross-sectional study, and that was a convenience sample.
This is a result “An opportunistic sample of 174 respondents returned the questionnaire (62.6% females, mean age 37.4, SD: 16.4) and was involved in the study.”
I believe that some more information on statistical analyses would be useful to the reader.
Discussion
Please add the point about health literacy versus information exchange and infodemic.
Author Response
Reviewer 3
This is an interesting research topic. Some points need to be reviewed to improve the manuscript.
Thank you for this comment! The following points were addressed as requested.
Abstract: It is not sufficiently clear. Please, introduction (01 line), and add aim, materials and methods and conclusion.
The abstract was amended as requested. The aim was more clearly indicated (row 14); material and methods are described at rows 17-20; a more focused conclusion on the link between results and applications to media communication for an effective management of emergencies and the increase of preventive behaviors was presented (rows 24-25).
Introduction: Regarding the introduction, consider that it is well structured and organized. It allows the reader to find the information necessary to promote understanding of the state of affairs. As a suggestion, consider presenting short, precise paragraphs. What is currently known? What does this article add?
Thank you for this appreciation! We introduced subsections in the long Section 1.2, and added a final section with Research aims and hypotheses, to address the added value of this study, as requested also by Reviewer 2.
It is not sufficiently clear that they refer to the adult population.
Thank you for this comment. We added more information about the sample (row 231-233).
Aim: It is not sufficiently clear in methods and abstract
We have better explained the aims of the study in the abstract (row 14) and in the new section Research aims and hypotheses.
Materials and Methods
I do, however, have some concerns/comments that I feel the authors need to address.
I believe that some more information on participants recruitment methods would be useful to the reader.
Was any kind of recompense for participation provided?
What are the inclusion and exclusion criteria? Who did the data collection?
Thank you for these comments. We added all the above information (rows 224-234).
Please include additional information regarding the survey or questionnaire used in the study and ensure that you have provided sufficient details that others could replicate the analyses. Who developed the questionnaire? Was there a pre-test?
Thank you for this comment. All the items were presented in the results of factor analyses. We added further information about the questionnaire and its development (rows 247-249).
I suggest Including that is a cross-sectional study, and that was a convenience sample.
Thank you for this suggestion. We added this information (row 191). We used the term "opportunistic" for the sample.
This is a result “An opportunistic sample of 174 respondents returned the questionnaire (62.6% females, mean age 37.4, SD: 16.4) and was involved in the study.”
We are just describing the sample, and further information was added.
I believe that some more information on statistical analyses would be useful to the reader.
Details are given in Section 2.3, but further suggestions are welcome.
Discussion
Please add the point about health literacy versus information exchange and infodemic.
Thank you for this suggestion. We added this relevant aspect in the discussion of the psychological mechanisms of media influence on healthy behaviors (rows 127-134).
Reviewer 4 Report
Undoubtedly, the authors have put a lot of work into their article. I am glad that they have taken up the current topic of the COVID-19 pandemic. Nevertheless, the text has raised a lot of questions, mainly concerning methodology and anchoring in the literature (State of Art). Some of them were formulated by the authors themselves in the last paragraph of point 4 (unfortunately, the submitted text does not contain verse numbers). My concern is, however, about several other issues.
1. Lack of theoretical foundation in the literature on media studies and media influence. First of all, reference should be made here to such theories as George Gerbner's cultivation theory, McCombs and Shaw's agenda-setting theory, L. Festinger's theory of cognitive dissonance. Without this, the theoretical part is incomplete. Embedding in these theories would also help to deepen the final conclusions;
2 The study itself refers to COVID-19, but unfortunately, it does not bring much novelty in relation to the hypotheses that have long been confirmed. Already the Janis & Fesbach study (1954) showed that too high a level of anxiety in health messages is ineffective. On the other hand, we have a number of studies which confirm that it is more likely to change behaviour where messages have aroused strong anxiety (Berkowitz&Cottingham 1960; Leventhal - several publications on the subject). Nevertheless, studies by Leventhal, Zimmerman & Gutmann 1984 have shown that information about health risks alone does not change behaviour, even if the information is received and understood. This is precisely the situation we have in the article under review;
3. comments on the methodology:
A. It is not clear whether the questionnaire was distributed on paper or in the form of a link, this is not apparent from the text.
B. There was practically no sampling. The research can only be treated as a survey. The students' relatives and friends are a completely accidental group, it is possible that some features were over-represented (e.g. the students' parents or friends could also have had higher education more often, could have lived closer to the capital). In fact, only age, gender and SD are indicated on the label. Therefore, it is not entirely clear who the respondents were - their potential proximity in relation to socio-demographic characteristics may have influenced the uniformity of results. That is why the results of these studies cannot be estimated for a larger population. In fact, the authors themselves are aware of this.
C. The text lacks information on how the 'three macro-themes of TV news' were created - how and of what type are these macro-themes? The respondents' attitude to the statements prepared by the researchers was examined. And this is interesting - where did these statements come from? What did the authors base their work on? Where did these statements come from and on what basis were they built in the questionnaire? Were these quotations from the media or did the researchers create them themselves?
D. Probably the questionnaire did not check whether the respondents used television at all, because it is mainly mentioned in the article, so why is there media influence in the title? Is it possible that television viewing in Italy was so high? Could the respondents also identify other sources of information (radio, press, portals, social media?).
E. What does "Media exposure" mean for the authors - How did they measure it? Or just the presence of information in the media?
As for the very interpretation of the study: The authors used factorial analysis. The analysis as such is ok. However, it seems to confirm conclusions that have long been proven by other researches - although not in relation to COVID-19.
Author Response
Reviewer 4
Undoubtedly, the authors have put a lot of work into their article. I am glad that they have taken up the current topic of the COVID-19 pandemic. Nevertheless, the text has raised a lot of questions, mainly concerning methodology and anchoring in the literature (State of Art). Some of them were formulated by the authors themselves in the last paragraph of point 4 (unfortunately, the submitted text does not contain verse numbers). My concern is, however, about several other issues.
Strange, we have verse numbers at the right of the manuscript… We hope they can be visible in the revised version.
- Lack of theoretical foundation in the literature on media studies and media influence. First of all, reference should be made here to such theories as George Gerbner's cultivation theory, McCombs and Shaw's agenda-setting theory, L. Festinger's theory of cognitive dissonance. Without this, the theoretical part is incomplete. Embedding in these theories would also help to deepen the final conclusions;
Thank you for this comment. We avoided considering all the relevant theories on media influence to focus only on specific mechanisms, but you are right. The inclusion of further literature would improve the discussion. We amended the paper accordingly with reference to both agenda-setting theory and cultivation theory (rows 114-122, and 391-392). The mechanism of cognitive dissonance, referring to a free choice of behavior from the person, seemed to be less relevant in a study on media influence.
2 The study itself refers to COVID-19, but unfortunately, it does not bring much novelty in relation to the hypotheses that have long been confirmed. Already the Janis & Fesbach study (1954) showed that too high a level of anxiety in health messages is ineffective. On the other hand, we have a number of studies which confirm that it is more likely to change behaviour where messages have aroused strong anxiety (Berkowitz&Cottingham 1960; Leventhal - several publications on the subject). Nevertheless, studies by Leventhal, Zimmerman & Gutmann 1984 have shown that information about health risks alone does not change behaviour, even if the information is received and understood. This is precisely the situation we have in the article under review;
The study investigated more than simply fear in influencing healthy behaviors. We discussed in the introduction the relevance of different processes – beyond receiving and understanding information - that make fear effective. Overall, people must know what to do, think they can, and believe it is effective (ref. 69-70). In this study, we found that fear increases healthy behaviors related to the prevention of COVID-19 when levels are moderate. The investigation of a new situation – COVID-19 - extends previous knowledge. Moreover, we found that moderate levels of fear work better together with other mechanisms. People have to think that the overall community is facing together the pandemic. We think that our findings, taken together add to previous literature. We discussed the integrated role of these mechanisms at rows 493-499.
- comments on the methodology:
A. It is not clear whether the questionnaire was distributed on paper or in the form of a link, this is not apparent from the text.
Thank you for suggesting this lack, we added the information in the text (row 225).
There was practically no sampling. The research can only be treated as a survey. The students' relatives and friends are a completely accidental group, it is possible that some features were over-represented (e.g. the students' parents or friends could also have had higher education more often, could have lived closer to the capital). In fact, only age, gender and SD are indicated on the label. Therefore, it is not entirely clear who the respondents were - their potential proximity in relation to socio-demographic characteristics may have influenced the uniformity of results. That is why the results of these studies cannot be estimated for a larger population. In fact, the authors themselves are aware of this.
As the Reviewer stated, we are aware of this. The world of COVID-19 has changed continuously, and TV news and discussions as well. That's why our aim was to give a picture of this world under clear conditions: at the start of the outbreak, for a population somewhat "distant" from the pandemic, and when communication was about specific themes. This choice was made in order to understand the psychological processes that could better link TV news, people's reactions, and healthy behaviors. Understanding prevention was the main objective of our investigation. We have tried to better discuss these points in the new subsection Research aims and hypotheses, as asked by Reviewer 2. Nonetheless, we are aware that the picture is not a movie, and thus we clearly commented on these aspects in the limitations of the study. The consequence of taking a picture under clear conditions is that we did not have much time (rows 230-231). That's why we opted for a convenience sample. We recognized also this limitation in the discussion (rows 479-480). Anyway, we added further information about the sample, showing that it has some variability (rows 231-233).
The text lacks information on how the 'three macro-themes of TV news' were created - how and of what type are these macro-themes?
Thank you for identifying this lack. We added this information.
The respondents' attitude to the statements prepared by the researchers was examined. And this is interesting - where did these statements come from? What did the authors base their work on? Where did these statements come from and on what basis were they built in the questionnaire? Were these quotations from the media or did the researchers create them themselves?
Attitude items were created by the authors of the study and pre-tested in terms of understandability by three colleagues. We added this information in the text (rows 247-249). Because of the need of taking a picture of people's reactions at specific conditions of the pandemic and news about it, unfortunately we did not have the time for a more accurate pre-testing.
Probably the questionnaire did not check whether the respondents used television at all, because it is mainly mentioned in the article, so why is there media influence in the title? Is it possible that television viewing in Italy was so high? Could the respondents also identify other sources of information (radio, press, portals, social media?).
We asked respondents to indicate how much time a day they viewed TV news and discussions about COVID-19. Our analysis in the study was thus about this aspect of media influence, and we included it in the title. Reviewer 4 is right when saying that information – and influence – can come from different sources. Our choice was to analyze TV news and discussions alone, in order to have a common background of information (for example, daily newscast). To give an example among other sources, social media can be very different, and controlling this variability very hard for us. However, we addressed this limitation, and the importance of future research on these aspects in the discussion (rows 488-489)
What does "Media exposure" mean for the authors - How did they measure it? Or just the presence of information in the media?
We considered exposure as the time reported by participants devoted to viewing TV news and discussions about COVID-19. We added this information more clearly at row 264.
As for the very interpretation of the study: The authors used factorial analysis. The analysis as such is ok. However, it seems to confirm conclusions that have long been proven by other researches - although not in relation to COVID-19.
We agree with this comment, even though the results also extended previous literature. The study showed the relevance of well-known psychological processes related to media exposure in influencing behavior. However, the novelty is in several aspects. First, we tested how these processes apply to the new emergency of COVID-19; second, we considered the combined contribution of different processes often investigated separately by previous literature; third, we found an interesting effect of perceiving social measures to increase healthy uses of public spaces (rows 493-499). Overall, these results can be important for the management of communication in future emergencies. We tried to better stress this aspect in the discussion (rows 499-504).
Round 2
Reviewer 1 Report
This manuscript results from the construction, usage and analysis of a questionnaire on the perceived media influence on healthy behavior on public spaces. It consists of three parts. In the first part, an overview is given of selected publications on the way COVID-19 influences society. In the second part the authors report on their own interview with students, friends and other relations. In the third part the reported data are analyzed and some conclusions are tentatively formulated.
The manuscript has been structured better than the original. However, the influence of structuring should not only be limited to the textual layout, but also to the conceptual structure. The lack of conceptual structure is typically visible in the tables and figures. The mass of unstructured data will confuse the reader. It will further block structuring partial observation into an overall view on the finding.
If the reported work is to be presented at a conference, you would expect at the start: a research question followed by 3 – 5 sub-questions. This vertical division could be independent, but preferably an abstraction level is used that allows re-using the pictorials. Once this is found, the paper will become transparent and the mass of measurement data can be moved to an appendix. This is the principle of experimental science, where knowledge is made from structuring labelled data. Without the structuring, it is just a soup.
The authors argue that they are working in a science niche where structuring is still not apparent. Therefore, they see their contribution as that of one of the many ants bringing food to the queen. I would gladly accept this if the authors can bring at least some glimmer of gold. I have tried to answer the question: would I refer to this manuscript in a discussion of data science in the health arena. I have failed to find this answer.
In all, the proposed paper has clearly improved but it is still missing the last step: giving it the reader something to remember. The presented data are still too coarse. The manuscript has no errors that can be pointed-out and repaired, but that is not enough to be published. It needs some more condensation and coloring to distinguish with just a measurement report. The work is typically not concluded and therefore the conclusion is not focused enough. This is not a question of a simple repair but rather proposing some possibilities, phrasing a hypothesis and finally approving/rejecting the outcome. In short: the contribution must be in taking a step. “Less can be more”.
Author Response
Dear Editor and Reviewers,
We would like to thank the Reviewers for their positive feedback about our revised manuscript. We are happy that all reviewers have appreciated and recognized our efforts in improving the paper.
We hope that the new version of the paper can be accepted for publication on IJERPH. Please consider that our replies to Reviewers’ comments are in italics.
Reviewer 1
This manuscript results from the construction, usage and analysis of a questionnaire on the perceived media influence on healthy behavior on public spaces. It consists of three parts. In the first part, an overview is given of selected publications on the way COVID-19 influences society. In the second part the authors report on their own interview with students, friends and other relations. In the third part the reported data are analyzed and some conclusions are tentatively formulated.
The manuscript has been structured better than the original. However, the influence of structuring should not only be limited to the textual layout, but also to the conceptual structure. The lack of conceptual structure is typically visible in the tables and figures. The mass of unstructured data will confuse the reader. It will further block structuring partial observation into an overall view on the finding.
If the reported work is to be presented at a conference, you would expect at the start: a research question followed by 3 – 5 sub-questions. This vertical division could be independent, but preferably an abstraction level is used that allows re-using the pictorials. Once this is found, the paper will become transparent and the mass of measurement data can be moved to an appendix. This is the principle of experimental science, where knowledge is made from structuring labelled data. Without the structuring, it is just a soup.
The authors argue that they are working in a science niche where structuring is still not apparent. Therefore, they see their contribution as that of one of the many ants bringing food to the queen. I would gladly accept this if the authors can bring at least some glimmer of gold. I have tried to answer the question: would I refer to this manuscript in a discussion of data science in the health arena. I have failed to find this answer.
In all, the proposed paper has clearly improved but it is still missing the last step: giving it the reader something to remember. The presented data are still too coarse. The manuscript has no errors that can be pointed-out and repaired, but that is not enough to be published. It needs some more condensation and coloring to distinguish with just a measurement report. The work is typically not concluded and therefore the conclusion is not focused enough. This is not a question of a simple repair but rather proposing some possibilities, phrasing a hypothesis and finally approving/rejecting the outcome. In short: the contribution must be in taking a step. “Less can be more”.
Authors’ reply: We are happy that the reviewer has recognized our great effort in improving the paper and that the manuscript has no errors that be pointed-out. We agree with the reviewer that sometimes the excess of data could distract readers from the main story. Consequently, we have decided to move tables of factorial analyses in the supplementary materials. In this way, readers can have in the paper only the tables central for conclusions about the hypotheses, following the suggestion that “less can be more”. At the same time, we have decided to leave factorial analyses in the main text, given that we strongly believe that sharing these data are important for readers.
Thank you for clarifying another point. We agree when the reviewer asked for an overall consideration merging together the different results of the study, and producing something which can clearly identify a contribution to remember. We added such a conclusive point in the conclusion (rows 532-541).
Reviewer 3 Report
I appreciate all of the work that you have put into this paper. The transformation from your original submission to the current version has made this a much stronger paper. However, I have a few remaining questions/comments/suggestions.
Aim: Move “cross-sectional study” to Study Design
I consider that the method could still be organized, and I suggest the sections:
- Materials and Method
2.1. Study Design - A cross-sectional study using an opportunistic sample method was conducted between ( ) and ( ) 2020 in Rome, Italy.
2.2 Participants: Include inclusion and exclusion criteria. You shouldn't be presenting results here.
2.3 Tools
2.4 Procedure or Data Collection
2.5 Statistical Analysis
2.6 Ethical Aspects
I can't entirely agree with this paragraph.
The first pertains to the use of convenience sampling and its cross-sectional nature.
Study Design is not a limitation.
Author Response
Dear Editor and Reviewers,
We would like to thank the Reviewers for their positive feedback about our revised manuscript. We are happy that all reviewers have appreciated and recognized our efforts in improving the paper.
We hope that the new version of the paper can be accepted for publication on IJERPH.
Reviewer 3
I appreciate all of the work that you have put into this paper. The transformation from your original submission to the current version has made this a much stronger paper. However, I have a few remaining questions/comments/suggestions.
We thank the reviewer for his/her comments which helped us to improve the manuscript.
Aim: Move “cross-sectional study” to Study Design
Authors’ reply: We agree with this useful suggestion.
I consider that the method could still be organized, and I suggest the sections:
- Materials and Method
2.1. Study Design - A cross-sectional study using an opportunistic sample method was conducted between ( ) and ( ) 2020 in Rome, Italy.
2.2 Participants: Include inclusion and exclusion criteria. You shouldn't be presenting results here.
2.3 Tools
2.4 Procedure or Data Collection
2.5 Statistical Analysis
2.6 Ethical Aspects
Authors’ reply: We have organized the method section following the useful suggestions given by the reviewer.
I can't entirely agree with this paragraph: “The first pertains to the use of convenience sampling and its cross-sectional nature”. Study Design is not a limitation.
Authors’ reply: We have deleted the study design from the limitations.
Reviewer 4 Report
Unfortunately, my version of the manuscript still doesn't contain the verse numbers :(
I would like to thank the authors for the clarifications and corrections they have made. I hope that my comments have helped to improve the article. I think it can be published as presented.
Author Response
Dear Editor and Reviewers,
We would like to thank the Reviewers for their positive feedback about our revised manuscript. We are happy that all reviewers have appreciated and recognized our efforts in improving the paper.
We hope that the new version of the paper can be accepted for publication on IJERPH. Please consider that our replies to Reviewers’ comments are in italics.
Reviewer 4
Unfortunately, my version of the manuscript still doesn't contain the verse numbers :(
I would like to thank the authors for the clarifications and corrections they have made. I hope that my comments have helped to improve the article. I think it can be published as presented.
Authors’ reply: Thank you very much for your precious suggestions that have helped us a lot to improve our manuscript!